# Influence of Particle Charge and Size Distribution on Triboelectric Separation—New Evidence Revealed by In Situ Particle Size Measurements

**Johann Landauer * and Petra Foerst**

Chair of Process Systems Engineering, TUM School of Life Sciences Weihenstephan, Technical University of Munich, Gregor-Mendel-Straße 4, 85354 Freising, Germany; petra.foerst@tum.de

* Correspondence: johann.landauer@tum.de; Tel.: +49-8161-71-5172

**Abstract:** Triboelectric charging is a potentially suitable tool for separating fine dry powders, but the charging process is not yet completely understood. Although physical descriptions of triboelectric charging have been proposed, these proposals generally assume the standard conditions of particles and surfaces without considering dispersity. To better understand the influence of particle charge on particle size distribution, we determined the in situ particle size in a protein–starch mixture injected into a separation chamber. The particle size distribution of the mixture was determined near the electrodes at different distances from the separation chamber inlet. The particle size decreased along both electrodes, indicating a higher protein than starch content near the electrodes. Moreover, the height distribution of the powder deposition and protein content along the electrodes were determined in further experiments, and the minimum charge of a particle that ensures its separation in a given region of the separation chamber was determined in a computational fluid dynamics simulation. According to the results, the charge on the particles is distributed and apparently independent of particle size.

**Keywords:** triboelectric separation; particle size distribution; particle charge; binary mixture; in situ particle size measurement; charge estimation

## 1. Introduction

Electrostatic effects were first recognized by the ancient Greek philosophers, who generated electricity by rubbing amber with fur. Thales of Miletus is often called the discoverer of the triboelectric effect [1,2]; however, this ancient observation has not been completely understood. Triboelectric charging of conductive materials is described by work function [3,4]. At conductor–insulator and insulator–insulator contacts, triboelectric charging has been described with "effective work function" [5,6], electron transfer [7], ion transfer [8,9], and material transfer [10,11]. Furthermore, contact charging is affected by environmental conditions such as humidity [12–14] and physical impact [15–17].

Triboelectric charging is undesired in process engineering because it interferes with pneumatic conveying [18,19], fluidized beds [20,21], mixing [22,23], and tablet pressing [24]. Moreover, it is a surface effect, indicating that particle surface plays a critical role. The known factors affecting triboelectric charging are particle area (indicated by particle size) [25–30], surface roughness [31–33], chemical composition [34], and elasticity (indicated by contact area) [35–38]; however, most experimental studies assumed uniform particles or contact surfaces. In most applications, the particles are not monodispersed and have no defined surface; however, the particles are dispersed in size, surface area, elasticity, crystallinity, and morphology.

To use triboelectric charging and subsequent separation as a tool to separate particles due to their chemical composition, surface morphology, crystallinity, or particle morphology, it is necessary to

understand the influence of particle size distribution or powder composition as well as the influence of non-ideal conditions. All these factors show the necessity of triboelectric separation experiments with real, but defined, powders (like starch and protein) in order to use triboelectric separation to enrich, e.g., protein in lupine flour [39] and to take into account further influencing factors. Furthermore, the use of a starch–protein mixture as a model substrate for triboelectric charging is anticipated to have a possible application to enrich protein out of cereals or legumes. This ability of triboelectric separation has been demonstrated [39–46].

Hitherto, lots of studies have been carried out with well-defined particles or with inhomogeneous and undefined organic systems. Supplementary to these findings, real powders with a defined chemical composition and dispersity in particle size should be investigated. The discussion of influencing factors, such as particle morphology or further particle properties, suggest that particle surface charge is affected by the particle size distribution, in turn influencing the triboelectric charging effect. As the particle charge strongly affects the subsequent separation step, particles with different charges become separated at different regions on the electrodes, depending on the flow profile in the separation chamber and the electric field strength. Therefore, we hypothesize that if particle size (as a proxy of surface area) influences the charging and the subsequent separation properties, then particles of different sizes will be separated at different regions on the electrodes.

## 2. Materials and Methods

### 2.1. Materials

Whey protein isolate (Davisco Foods International, Le Sueur, MN, USA) with a protein content of 97.6 wt % was ground as that described in Landauer et al. [47]. Barley starch (Altia, Finland) with a starch content of 97.0 wt % was narrowed in particle size distribution in a wheel classifier (ATP 50, Hosokawa Alpine, Augsburg, Germany) under the conditions described in Landauer et al. [47].

### 2.2. Methods

#### 2.2.1. Separation Setup

The simple experimental setup, originally demonstrated by Landauer et al. [47,48], comprises of an exchangeable charging section and a rectangular separation chamber. The dispersion of powders added to the gas flow is facilitated by a Venturi nozzle. The charging tube (of diameter 10 mm and length 230 mm) was composed of polytetrafluoroethylene (PTFE). An electrical field strength of 109 V/m was applied to the parallel-plate capacitor in a rectangular separation chamber (46 mm × 52 mm × 400 mm). The protein contents of the binary protein–starch mixtures were varied as 15, 35, and 45 wt %, and were determined as described in [48]. Briefly, the powder was dispersed in NaCl buffer (pH 7, 0.15 M) and the protein concentration was photometrically determined at 280 nm. To measure the protein content along the electrodes, the powder was sampled in three colored areas (see Figure 1a).

The amount of particles separated along the electrodes was determined by measuring the height of the separated powder. The measuring points are shown in Figure 1a and marked with gray circles. The powder deposition height was measured homogenously along each electrode in order to get a topography. Note that the powder height was determined using a micrometer screw (according to DIN 863-1:2017-02). The mean was calculated from the results of three independent separation experiments ($n = 3$). Error bars indicate the confidence intervals of the Student's *t*-test with an $\alpha = 0.05$ significance level.

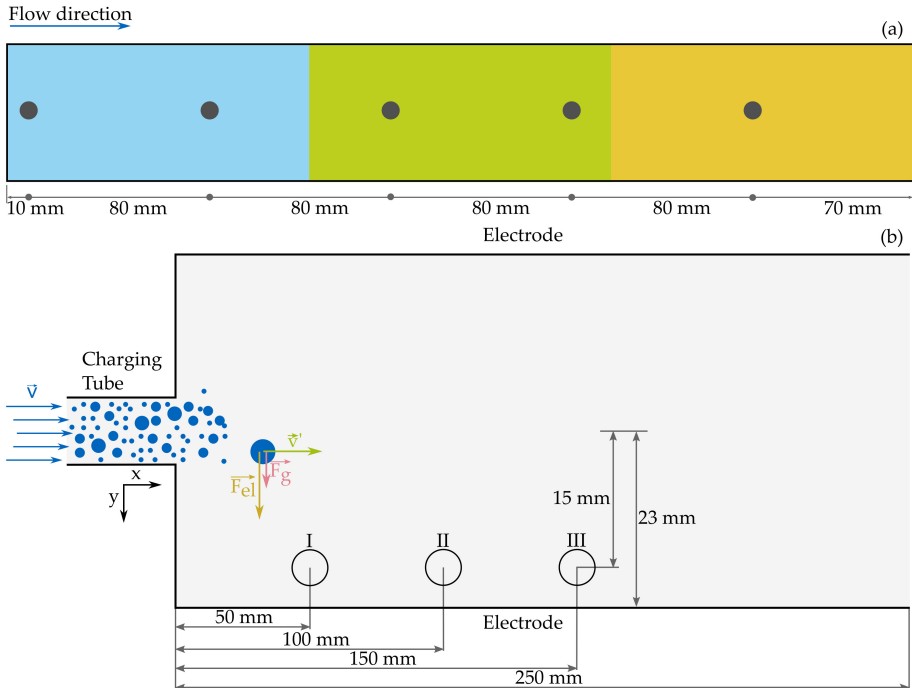

**Figure 1.** (**a**) Schematic of the sampling points along the electrodes. The powder deposition height on the electrode was measured at the points marked by gray circles. Colored areas mark the areas of powder sampling along the electrode. (**b**) Schematic of the separation chamber and the charging tube. The in situ particle size near the electrodes was measured at positions I, II, and III. Measurements near the anode and cathode were enabled by switching the polarity of the electrical field. The electric force $F_{el}$ and weight force $F_g$ acting on each particle in the separation chamber are visualized.

### 2.2.2. In Situ Particle Size Analysis

To investigate whether the particles agglomerate along the charging tube, we analyzed the in situ particle size distributions along the charging tube. For this purpose, parts of the charging and separation setup were installed in the measuring gap of a HELOS laser diffraction system (Sympatec, Clausthal-Zellerfeld, Germany). The charging and dispersing setup has been described in previous studies [47,48]. In the charging section, the particle size distributions were determined at the outlet of the Venturi nozzle (inlet of the charging section) and at the outlet of the charging tube. In the separation chamber, the particle size distribution was measured as a function of length. The schematic in Figure 1b shows the measuring positions I, II, and III along the electrodes in the separation chamber. The measuring points were chosen to be close to the electrodes and in the first half of the separation chamber. Due to the triboelectric separation, the particle concentration is decreasing along the separation chamber. Thus, the particle concentration, which is required to determine the particle size distribution, was not accessible. The particle size distributions on the anode and the cathode were obtained by switching the polarity of the electrical field with an electrical field strength of 217.4 kV/m. The mean of six independent separation experiments ($n = 6$) was calculated and the error is indicated by the confidence intervals of the Student's *t*-test with an $\alpha = 0.05$ significance level using error bars.

### 2.2.3. Flow Simulation and Estimation of the Particle Charge

The change in cross section between the charging tube and the separation chamber is very rigorous. The flow profile in the separation chamber, which might affect the separation characteristics and the particle size distribution along the electrodes (Figure 1b), was investigated in a computational fluid dynamics (CFD) simulation (ANSYS Fluent, version: 17.0, supplier: Ansys, Inc., Canonsburg, PA,

USA) of a realizable k–ε model. The particle motion in the separation chamber was visualized by tracking particles in the flow simulation. The inserted spherical particles followed a Weibull size distribution with a minimum, mean, and maximum (measured in the initial particle size distribution) of 1, 16, and 40 μm, respectively. The powder density was considered as the mean of the true density (1465 kg/m$^3$), which was measured by a gas pycnometer (Accupyc 1330, Micromeritics Instrument Corp., Norcross, GA, USA). The minimum charge at which the particle will be deflected in the measuring region was determined by simulating the in situ particle size distribution at different gravity levels (emulating different particle charges). The Coulomb force aligns with the weight force, as shown in Figure 1b. The absolute value of the charge $q$ of the particles is estimated as follows:

$$q = \frac{x_3}{6E}\pi\rho_s g(n-1) \tag{1}$$

where $x$ is the mean particle size, $\rho_s$ is the true density, $\vec{E}$ is the electrical field, $\vec{g}$ is gravity, and the scaling factor $n$ is the increase in particle charge. The scaling factor in the simulation was varied between 1 and 44. Note that the polarity of the charge depends on the electrical field's polarity.

## 3. Results

### 3.1. Particle Size Distribution

#### 3.1.1. Agglomeration within the Charging Tube

Figure 2 shows the volume–weight density distributions at the Venturi nozzle outlet (panel a) and at the outlet of the charging tube (panel b) in the 15 and 30 wt % powders. In both particle size distributions, the particle size decreased with increasing initial protein content. The particle size distributions were similar at the outlets of the nozzle and the charging tube. The mean particle size (peak position) and the maximum particle size are the same at the tube inlet and the tube outlet. By comparing the distribution of finer particles, an increase in finer particles is visible. Thus, a dispersion along the charging tube is measured. The reason for this dispersion could be the high particle–particle collision number within the charging tube, due to the high turbulence [47]. The results in Figure 2 indicate breaking up particle agglomerates of fine particles during the charging step that could promote electrostatic separation. Contrarily, no electrostatic agglomeration that could impair electrostatic separation is observed.

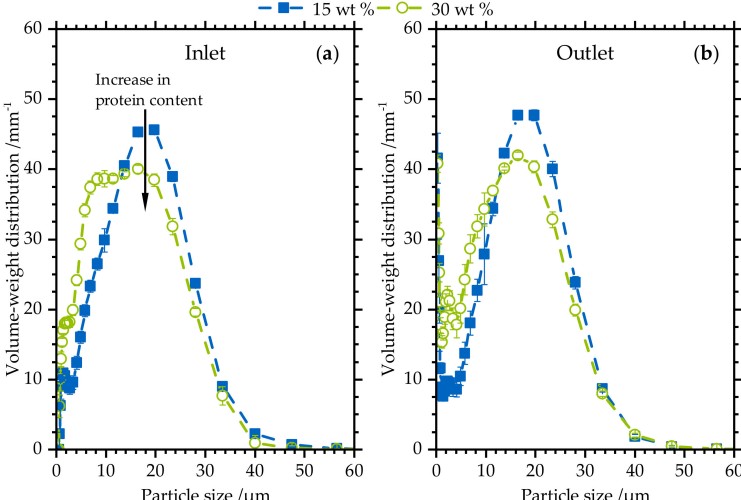

**Figure 2.** Volume–weight density distribution at (**a**) the Venturi nozzle outlet and (**b**) the outlet of the charging tube. Increasing the initial protein content (from 15 to 30 wt %) refined the particle size distribution. The distributions at the nozzle and tube ends are not obviously different.

### 3.1.2. Particle Size Distribution along the Electrodes

　　Figure 3 shows the volume–weight density distributions of the powder close to the cathode (a) and the anode (b) in measuring regions I, II, and III (Figure 1b). Increasing the initial protein content refined the particle size distributions at both the cathode and anode, as evidenced by the higher peak at ~6 μm in the 30 wt % compared to the 15 wt % distribution. This higher peak indicates a higher amount of finer protein particles (cf. Figure 2). In the sample with an initial protein content of 15 wt %, the particle size decreased along the investigated regions I, II, and III (note the lower peak height at 16 μm than that at 6 μm). This stepwise decrease in particle size was observed along both the cathode and the anode, as well as in the sample with higher initial protein content (30 wt %). The peak increases from 15 to 30 wt % are more clearly observed at 6 μm compared to 16 μm because increasing the protein content increases the amount of smaller particles. Comparing the particle size distributions at the cathode and the anode, the particles were finer on the cathode regardless of the initial protein content. These results suggest a higher protein content on the cathode (cf. Figure 2). The protein content of the separated powder on the cathode and the anode is approximately 80 and 2.5 wt %, respectively. Thus, protein is enriched on the cathode and starch is enriched on the anode [47,48]. However, the enhancement of finer protein particles near the cathode cannot be correlated with the protein content because the used protein powder is finer than the starch powder. Nevertheless, the particle size distributions at each measuring position in the separation chamber depended on the initial protein content. Thus, the particle size is influenced by the polarity of the electric field, the distance from the charging tube outlet, and (most strongly) the initial protein content. The region in which the particles separate plays a subordinate role on the particle size distribution. Furthermore, the particle size distributions on the anode and cathode resembled the initial distributions determined at the outlets of the Venturi nozzle and the tube.

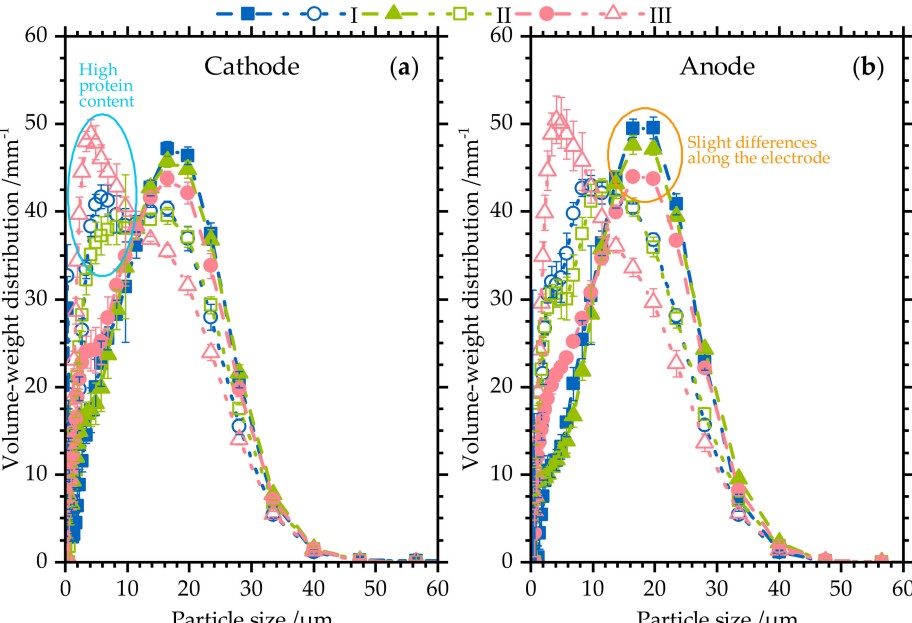

**Figure 3.** Volume–weight density distributions recorded near the cathode (**a**) and the anode (**b**) in regions I, II, and III. Closed and open symbols denote initial protein contents of 15 and 30 wt %, respectively. Increasing the initial protein content reduces the particle size at both anode and cathode. The particle size distributions differ in the three measuring regions.

### 3.2. Powder on the Electrodes

#### 3.2.1. Powder Height

Figure 4 shows the powder height along the cathode and the anode. On the cathode, the distribution of powder was approximately homogeneous along the electrode. The powder height varied most extensively at the second measuring point, and was least variable at the first and fourth measuring points. This significant but extremely low variation should not be overinterpreted; however, the powder height severely decreased along the anode. The powder height was constant at the first two measuring points, and then dropped to zero over the next two measurement points. Thus, the powder heights on the cathode and the anode exhibited very different profiles, suggesting different charges of the particles separated on the two electrodes. In particular, the negatively charged particles exhibited a higher net charge than the positively charged particles.

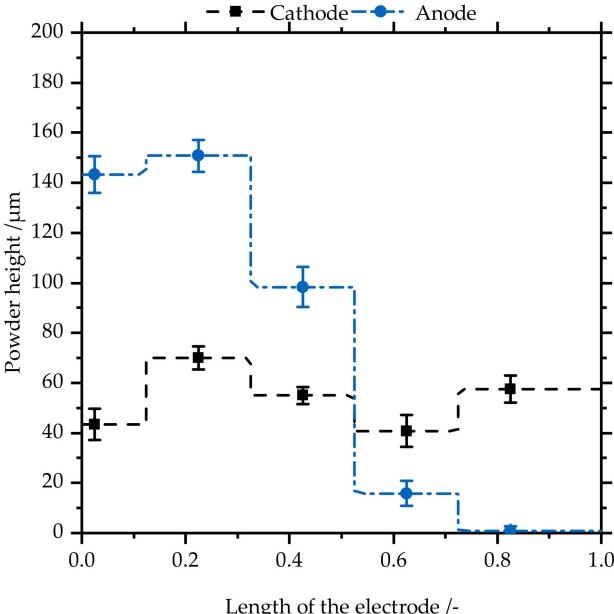

**Figure 4.** Powder height along the cathode and the anode. Powder height is approximately constant on the cathode, but mostly separates over the first half of the anode.

#### 3.2.2. Protein Content

Figure 5 shows the protein content on the cathode in the three measurement areas at initial protein contents of 15 and 30 wt %. The protein content was consistently higher for the sample with the higher initial protein content. Independently of the initial protein content, the protein content increased in the second area (relative to the first area). In the third area, the protein content decreased at the initial protein content of 15 wt %, but remained high at the higher initial protein content. If we compare the protein content with the powder height, the two quantities are apparently independent because the powder height was approximately constant along the electrode, whereas the protein content extensively varied.

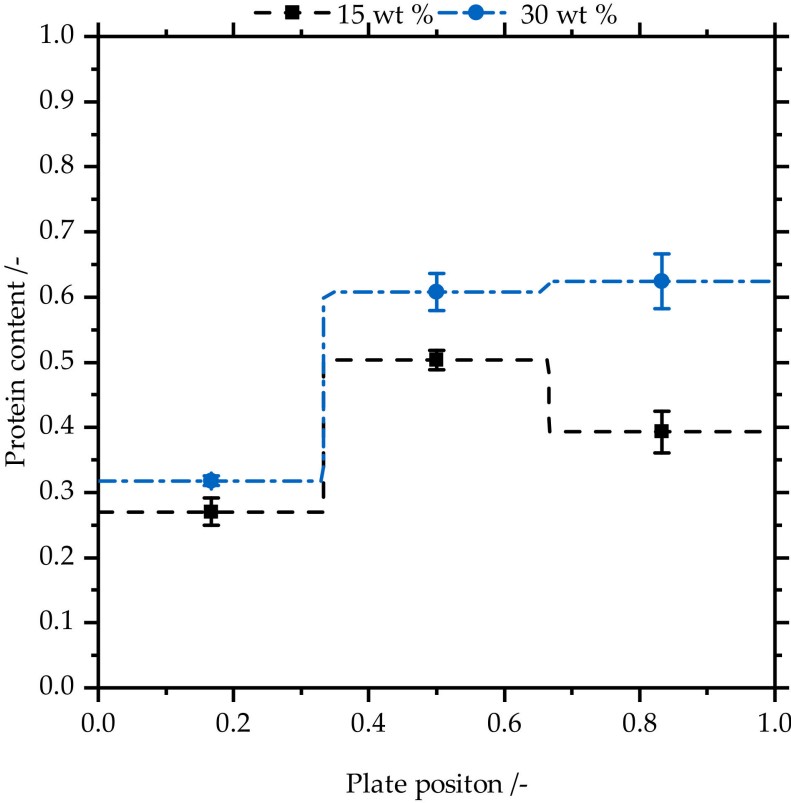

**Figure 5.** Protein content on the cathode in three different measurement areas for initial protein contents of 15 and 30 wt %. The protein content increases form the first to the second area regardless of initial protein content. In the third area, the protein content decreases (15 wt %) or remains the same (30 wt %).

*3.3. Estimation of the Charge Correlated with the In Situ Particle Size Distribution*

To estimate the minimum charge at which particles will separate in the separation chamber (enabling an in situ particle size analysis), the particles were tracked in a CFD study. Figure 6 shows the trajectories of spherical particles with different accelerations (varied by changing the electrical force in Equation (1)). In a homogenous electrical field, the net charge of the particles is a multiple of the elementary charge. Uncharged particles might be undetectable in every measuring region. Particles with a net charge of $1.45 \times 10^3$ $q_e$ can be detected in regions II and III, whereas those with charges of $7.26 \times 10^3$ $q_e$ and $1.16 \times 10^4$ $q_e$ might be measurable only in region II. Particles with a net charge of $3.77 \times 10^4$ $q_e$ and higher are visible in region I. When generating Figure 6 and calculating the associated particle charge, we assumed spherical particles with a mean diameter of 16 μm corresponding to the mean particle size of the powders used for the experiments. According to Equation (1), the particle size affects both the charge on a single particle and the particle trajectories. In all cases, varying the particle size only slightly affected the trajectories.

The background of Figure 6 shows the velocity profile in the separation chamber. The profile shows a jet at the charging tube outlet followed by homogeneity. The jet formed at the outlet of the tube affected the particle trajectories considerably. Regardless of their net charge, the particles remained within the jet to ~100 mm from the outlet. Then, they lost speed and were deflected toward the electrode by the Coulomb force. Thus, the simulation visualized the influence of the particle net charge on the particle trajectories within a complex velocity profile. Observing these particle trajectories, we can understand how particles might be charged to ensure their separation in the measuring regions of in situ particle size analyses.

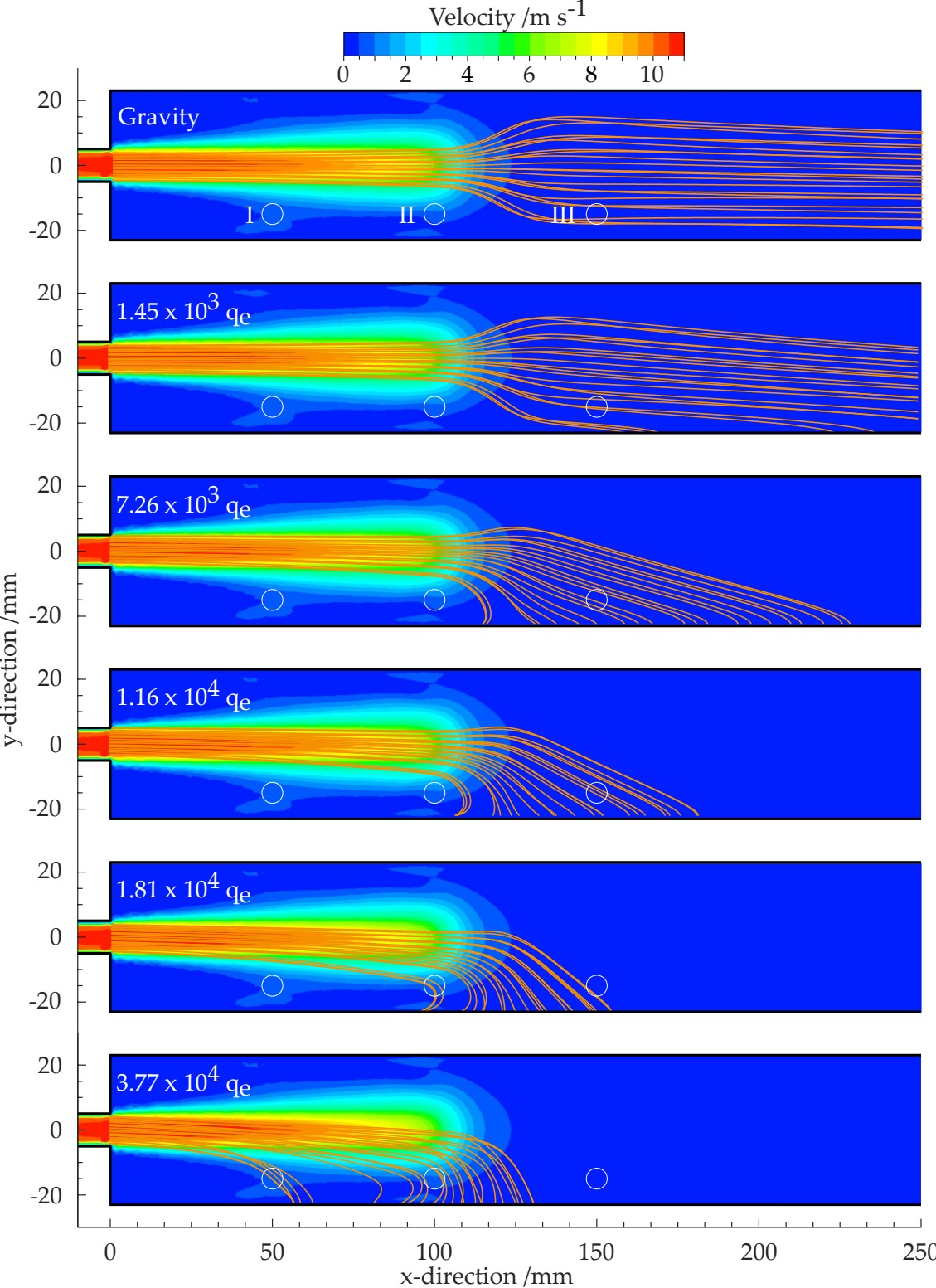

**Figure 6.** Trajectories of 16 μm diameter spherical particles with different particle charges (multiples of the elementary charge calculated by Equation (1)). The measuring regions I, II, and III of the in situ particle size distribution are indicated by the white open circles. Depending on their net charges, certain particles are not detectable in every measuring region. The background visualizes the velocity profile. The jet formed at the outlet of the charging tube is clearly visible.

## 4. Discussion

To use triboelectric separation as a tailor-made particle separation tool, one must separate the particles by their specific chargeabilities. Accordingly, it is necessary to disperse the particles before the charging step and avoid their agglomeration during the charging step. The selected setup enables the appropriate conditions for dispersal and aggregation prevention (Figure 2). Hence, detailed investigations of the separation step are required to establish triboelectric separation as an industrial separation technique.

Assuming that the charge distribution of fine particles is sourced from the triboelectric charging of the particles and that the charge distribution also possibly depends on the particle size and the chemical composition [34], the particle size distributions were determined at different locations close to the electrodes (Figure 3). As expected, the particle size distribution was coarser on the anode than on the cathode, because increasing the protein content refined the particle size distribution (Figure 2) and the protein was enriched on the cathode [47,48]. Moreover, along the measuring regions close to the electrodes, the decreased particle size accompanying the refined particles was demonstrated for different initial protein contents. These results were identical on the cathode and the anode, suggesting (as a first hint) that the net charges of the particles after triboelectric charging are independently distributed of the particle sizes.

The local distribution of the separated powder on the electrodes indicates the strength of the particle charges because particles with a higher and lower net charge are separated at the inlet and the near-outlet of the electrode, respectively. The powder height profiles on the cathode and the anode exhibited very different characteristics (Figure 4). The powder was dispersed almost homogeneously on the cathode but was separated close to the inlet on the anode. The absence of powder at the anode outlet indicates that the negative particles were more highly charged than the positive ones. The same results were reported in single-particle charge measurements [49]. This result further indicates independent distributions of the particle charges and sizes because the particle size distributions were similar on the cathode and anode (Figure 3). Furthermore, the homogeneously distributed powder exhibited a distributed protein content with a peak in the middle of the electrode at 15 wt % initial protein content, and level peaks in the second and third parts of the electrode at 30 wt % initial protein content (Figure 5). This suggests a lower net charge of protein particles than of starch particles (Figure 6). These results are underpinned by the decreased particle size (higher protein content) along the cathode than along the anode (Figure 3). The particle trajectories were affected by the inhomogeneous flow profile in the separation chamber; however, in the flow simulation, they were predominantly influenced by the charge. Moreover, they showed a charge-dependent separation region. To summarize, the binary powder mixture with a polydispersed particle size distribution showed no clear relationship between particle size and particle charge in the separation region. These results contradict previous studies, which reported that smaller particles are predominantly negatively charged [25–30]. The results support an effect of particle size on triboelectric charging, but no clear tendency was found regarding the fine and coarse particles. Thus, the hypothesis of this study, i.e., that particle size distribution (as a measure of surface area) plays a major role in triboelectric charging and subsequent separation, is questionable. Indeed, there is a dependence of particle size along the electrodes, but the results show a more complex connection between the particle material and particle size.

## 5. Conclusions

The dispersing and agglomeration characteristics of powders with different initial protein contents were consistent along the charging tube. The in situ particle size measurements were consistent at different regions in the separation camber. After estimating the minimum charge for particle separation, it was found that large charge differences were required for separation in every measuring region of the chamber. This wide charge distribution might lead to different separation regions of the particles, as indicated by the roughly homogeneous powder height on the cathode and the steep decrease in powder height on the anode. These results show a complex dependency of triboelectric charging and

subsequent separation on the size and material of the particles. As the mechanism of triboelectric or contact charging has not been accurately determined, determining the primary influencing factors is very challenging. The present results indicate the high complexity of triboelectric charging and indicate that particle size is not a highly important factor in triboelectric separation but affects the triboelectric charging through surface-area differences.

**Author Contributions:** Conceptualization, J.L.; methodology, J.L.; data curation, J.L.; writing—original draft preparation, J.L.; writing—review and editing, J.L. and P.F.; visualization, J.L.; supervision, P.F.

**Funding:** This research received no external funding.

**Acknowledgments:** The authors would like to thank Lukas Hans for help with performing in situ particle size measurement and Heiko Briesen for the possibility to carry out this study.

**Conflicts of Interest:** The authors declare no conflict of interest.

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
