# Peer review of "Influence of Particle Charge and Size Distribution on Triboelectric Separation—New Evidence Revealed by In Situ Particle Size Measurements"

_processes, doi:10.3390/pr7060381_

Round 1
Reviewer 1 Report
The authors got conclusions that the dependency between particle charge and particle size is complicated and not obviously linked, based on the set up mentioned in this and their previous (doi:10.1016/j.apt.2019.03.006.) studies. The study also showed that small particles are not predominantly negatively charged. The experimental was properly conducted and the results are worth to be published. Please consider the following revision suggestions.
1. In Figure 1, given the chamber is not vacuumed to my best knowledge, the gas flow in charging tube and in separation chamber should be different because of the difference of the cross-section area. Thus, it is not proper to assume the particle would maintain the same horizontal flying speed (or use the same ‘v’).
2. It could be helpful to clarify somewhere in the text about which electrode (anode, cathode) would mainly collect which type of particles (protein, starch), due to which charge (positive, negative), like shown in the authors previous paper (doi:10.1016/j.apt.2019.03.006.)
3. Figure 3 is difficult to comprehend due to many variables and much overlapped data. Could the authors make it a little bit more convenient to read such as put a ‘cathode’ in the figure after mark ‘(a)’ and an ‘anode’ after mark ‘(b)’, and point out some of those differences with markers that were mentioned in the text, as that in the Figure 2.
Author Response
Response to Reviewer 1 Comments
The authors have presented in-situ measurement of protein–starch mixture particle size distribution in order to better understand the influence of particle charge on particle size distribution. It has been done by both numerical simulation and experiments. This is a very interesting work and has technical merit. However, I have some concern about the content and presentation which must be addressed before acceptance. Please see my comments below:
Thank you for this encouraging comment.
Point 1: In Figure 1, given the chamber is not vacuumed to my best knowledge, the gas flow in charging tube and in separation chamber should be different because of the difference of the cross-section area. Thus, it is not proper to assume the particle would maintain the same horizontal flying speed (or use the same ‘v’).
Response 1: Thank you for this hint. Indeed, the velocity of the particles is differing according their position. The estimation of the velocity in the charging tube due to cross sectional area is supported by numerical flow simulations. The name of the particle velocity was changed to v’.
Point 2: It could be helpful to clarify somewhere in the text about which electrode (anode, cathode) would mainly collect which type of particles (protein, starch), due to which charge (positive, negative), like shown in the authors previous paper (doi:10.1016/j.apt.2019.03.006.).
Response 2: Thank you for your advice. In order to emphasize on which electrode protein and starch are predominately separated, the following changes were made (line 157-159):
The protein content of the separated powder on the cathode and the anode is approximately 80 wt.% and 2.5 wt.%, respectively. Thus, protein is enriched on the cathode and starch is enriched on the anode [47,48].
Point 3: Figure 3 is difficult to comprehend due to many variables and much overlapped data. Could the authors make it a little bit more convenient to read such as put a ‘cathode’ in the figure after mark ‘(a)’ and an ‘anode’ after mark ‘(b)’, and point out some of those differences with markers that were mentioned in the text, as that in the Figure 2.
Response 2: Thank you for this advice. To make the content of Figure 3 more accessible and convenient to read, ‘anode’ and ‘cathode’ labels were included and the main differences were marked.

Reviewer 2 Report
The authors have presented in-situ measurement of protein–starch mixture particle size distribution in order to better understand the influence of particle charge on particle size distribution. It has been done by both numerical simulation and experiments. This is a very interesting work and has technical merit. However, I have some concern about the content and presentation which must be addressed before acceptance. Please see my comments below:
1. Page 1, Title: The title must be rephrased.
2. Page 1, Abstract: Please remove the numbering in the Abstract.
3. Page 1-2, Introduction: The introduction should include more background information including methodologies, significance, drawbacks, as well as how the content of the manuscript and/or proposed method(s) would enhance the significance over the drawbacks identified.
4. Page 2, 2.1 Materials: More information about the materials must be included. Their properties and well as reason for choosing these materials in this work etc.
5. Page 2, 2.2.1. Separation setup: Require more information about the sample areas and the reason for choosing those sample areas. I would also recommend adding the schematic block diagram of the complete experimental setup.
6. ‘2.2.2. Powder height on the electrodes’ could be merged into ‘2.2.1. Separation setup’.
7. Page 4, 2.2.5. Statistics: More details are expected in this section. It could also be merged into ‘2.2.4. Flow simulation and estimation of the particle charge’
8. Page 4, 3.1.1. Agglomeration within the charging tube: Please explain the context of the result presented in Fig. 2, especially the reason of the variation of results.
9. Page 7, 3.3. Estimation of the charge correlated with the in-situ particle size distribution: What might be the best reason to assume the particle size to be 16 μm?
10. Page 8, 4. Discussion: “Hence, further investigations of the separation step are required.”
- Any suggestion would be of interest to the reader about the type of further investigation.
11. Require English correction throughout the manuscript.
Author Response
Response to Reviewer 2 Comments
The authors have presented in-situ measurement of protein–starch mixture particle size distribution in order to better understand the influence of particle charge on particle size distribution. It has been done by both numerical simulation and experiments. This is a very interesting work and has technical merit. However, I have some concern about the content and presentation which must be addressed before acceptance. Please see my comments below:
Thank you for this encouraging comment.
Point 1: Page 1, Title: The title must be rephrased.
Response 1: Thank you for this hint. We tried to rephrase the title, but the possible alternatives could not convey the main massage of our study. In the title we anticipate the main findings of our study. This seems to be a bit uncommon, but we believe that the main findings in the title can arise interest in the novel experimental setup. If you can give us a more detailed advice to improve the title, we are happy to reconsider our choice.
Point 2: Page 1, Abstract: Please remove the numbering in the Abstract.
Response 2: Thank you for your hint. Probably we misinterpreted the guidelines for authors. Thus, the numbers in the Abstract were removed.
Point 3: Page 1-2, Introduction: The introduction should include more background information including methodologies, significance, drawbacks, as well as how the content of the manuscript and/or proposed method(s) would enhance the significance over the drawbacks identified.
Response 3: Thank you for this hint. To categorize the study in a more detailed way in the current literature and to emphasize the necessity of the investigation of the influence of particle size distribution on triboelectric charging and subsequent separation, the following changes were made (line 43-56):
To use triboelectric charging and subsequent separation as a tool to separate particles due to their chemical composition, surface morphology, crystallinity, or particle morphology, it is necessary to understand the influence of particle size distribution or powder composition as well as the influence of non-ideal conditions. All these factors show the necessity of triboelectric separation experiments with real, but defined powders (like starch and protein) in order to use triboelectric separation to enrich, e.g., protein in lupine flour [39] and to take into account the further influencing factors. Furthermore, the use of a starch-protein mixture as a model substrate for triboelectric charging should anticipate a possible application to enrich protein out of cereals or legumes. This ability of triboelectric separation was demonstrated [39–46].
Hitherto, lots of studies were carried out with well-defined particles or with inhomogeneous and undefined organic systems. Supplementary to these findings, Real powders with a defined chemical composition and dispersity in particle size should be investigated. The discussion of influencing factors such as particle morphology or further particle properties suggest that particle surface charge is affected by the particle size distribution, in turn influencing the triboelectric charging effect.
Point 4: Page 2, 2.1 Materials: More information about the materials must be included. Their properties and well as reason for choosing these materials in this work etc.
Response 4: Thank you for your advice. As marked out in the changes to question 3 the mixture of protein and starch was chosen as a simple but defined real powder system in order to investigate the influence of particle size distribution on triboelectric charging. Besides this, triboelectric separation might be a suitable way to enrich proteins form cereals and legumes to improve their nutritional properties. Following changes were made:
All these factors show the necessity of triboelectric separation experiments with real, but defined powders (like starch and protein) in order to use triboelectric separation to enrich e.g. protein in lupine flour [39] and to emphasize the further influencing factors. Furthermore, the use of a starch-protein mixture as a model substrate for triboelectric charging should anticipate a possible application to enrich protein out of cereals or legumes. This ability of triboelectric separation was demonstrated [39–46].
Point 5: Page 2, 2.2.1. Separation setup: Require more information about the sample areas and the reason for choosing those sample areas. I would also recommend adding the schematic block diagram of the complete experimental setup.
Response 5: Thank you for this hint. The sample areas are shown in Figure 1. In Figure, 1a the measuring points of the powder-deposition height (grey circles) and the regions for powder sampling along the electrode (colored areas) for protein content determination. The sample areas in for powder-deposition height are dispersed homogenous along the electrodes. For powder sampling, the electrode is divided into three regions, because analysis of the protein content requires approximately the amount of powder collected in one area. The measuring position of in-situ particle size distribution were chosen under the requirements that particle size distribution is measured in proximity to the electrodes and in the first half of the separation chamber, because the in the second half of the camber the necessary particle concentration is not achieved (cf. Figure 4). To specify why choosing the sample areas following changes were made (line 77, 81, 95):
To measure the protein content along the electrodes, the powder was sampled in three colored areas (see Figure 1a).
The measuring points are shown in Figure 1a and marked with gray circles. The powder-deposition height was measured homogenously along each electrode in order to get a topography.
The measuring points were chosen to be close to the electrodes and in the first half of the separation chamber. Due to the triboelectric separation, the particle concentration is decreasing along the separation chamber. Thus, the particle concentration which is required to determine the particle size distribution was not accessible.
Point 6: ‘2.2.2. Powder height on the electrodes’ could be merged into ‘2.2.1. Separation setup’.
Response 6: Thank you for this suggestion. As you proposed, ‘Powder height on the electrodes’ was merged with ‘Separation Setup’.
Point 7: Page 4, 2.2.5. Statistics: More details are expected in this section. It could also be merged into ‘2.2.4. Flow simulation and estimation of the particle charge’
Response 7: Thank you for this advice. To improve the structure of the manuscript information on statistics is included in section 2.2.1 and 2.2.2. Additionally, the following changes were made (line 84, 100):
The mean was calculated from the results of three independent separation experiments (n = 3). Error bars indicate the confidence intervals of the Student’s t-test with an α = 0.05 significance level.
The mean of six independent separation experiments (n = 6) was calculated and the error is indicated by the confidence intervals of the Student’s t-test with an α = 0.05 significance level using error bars.
Point 8: Page 4, 3.1.1. Agglomeration within the charging tube: Please explain the context of the result presented in Fig. 2, especially the reason of the variation of results.
Response 8: Thank you for your advice. The agglomeration within the charging tube form the basis of in-situ particle size measurement. If there is electrostatic agglomeration along the charging step, powder separation is limited. Along the charging tube, no electrostatic agglomeration was determined, because no increase of the mean particle size (the positon of the peak) and the maximum particle size were found. These findings were made for two different mixtures varying in the protein content. At a higher initial protein content, a dispersion along the charging tube is visible indicated by a higher amount of finer particles. This dispersion could be the result of a high particle-particle collision number within the charging tube. In order clarify the reasons for the variation in the particle size distributions, following changes were made (line 132):
The mean particle size (peak position) and the maximum particle size are the same at the tube inlet and the tube outlet. By comparing the distribution of finer particles, an increase in finer particles is visible. Thus, a dispersion along the charging tube is measured. The reason for this dispersion could be the high particle-particle collision number within the charging tube due to the high turbulence [47]. The results in Figure 2 indicates breaking up particle agglomerates of fine particles during the charging step that could promote electrostatic separation. Contrary, no electrostatic agglomeration that could impair electrostatic separation could be observed.
Point 9: Page 7, 3.3. Estimation of the charge correlated with the in-situ particle size distribution: What might be the best reason to assume the particle size to be 16 μm?
Response 9: Thank you for this hint. In order to plot results of the simulation study, the mean particle size of the used powders was chosen, because the mean particle size is commonly used to describe the narrow particle size distributions. In order to explain the reason for choosing the particle size of 16 µm, following changes were made (line 206):
When generating Figure 6 and calculating the associated particle charge, we assumed spherical particles with a mean diameter of 16 µm corresponding to the mean particle size of the powders used for the experiments.
Point 10: Page 8, 4. Discussion: “Hence, further investigations of the separation step are required.”- Any suggestion would be of interest to the reader about the type of further investigation.
Response 10: Thank you for this advice. We agree with your advice that the reader might expect which further investigations have to be done. We wanted to arouse interest of the reader to the “further investigations” we presented in the current paper. To better guide the reader and to focus her curiosity on the further investigation we did in the current study, following was changed (line 228):
Hence, detailed investigations of the separation step are required to establish triboelectric separation as an industrial separation technique.
Point 11: Require English correction throughout the manuscript.
Response 11: Thank you for this comment. English is corrected throughout the manuscript.

Round 2
Reviewer 2 Report
The authors have responded to my comments pretty well and revised the manuscript accordingly. I recommend accept the manuscript with the following correction:
Current title looks like a statement. By rephrasing I meant that the sentence should be modified with the same meaning. It might be 'Influence of particle charge and size distribution in triboelectric separation by in-situ particle size measurement' or something similar to that.
Author Response
Point 1: Current title looks like a statement. By rephrasing I meant that the sentence should be modified with the same meaning. It might be 'Influence of particle charge and size distribution in triboelectric separation by in-situ particle size measurement' or something similar to that.
Response: Thank you for your comment. We rephrased the title in order to avoid the statement:
Influence of particle charge and size distribution on triboelectric separation – New evidence revealed by in-situ particle size measurements